# Price Transmission Analysis of the International Soybean Market in a Trade War Context

Gustavo Barboza Martignone [1], Karl Behrendt [2] and Dimitrios Paparas [3,*]

1   Food, Land and Agribusiness Management Department, Harper Adams University, Newport TF10 8NB, UK
2   Agri-Tech Economic Modelling at Harper Adams University, Newport TF10 8NB, UK
3   Economics at Harper Adams University, Newport TF10 8NB, UK
*   Correspondence: dpaparas@harper-adams.ac.uk

**Abstract:** This study analysed the dynamics of the international soybean market using econometric techniques and economic models to study the impacts of the US–China trade war. It considered the analysis of "spatial" (horizontal) price transmission during an approximately ten-year period from September 2009 to May 2019 using monthly time-series data. The research focused on the leaders in the international soybean market, namely, China, the USA, the EU, Brazil and Argentina. Several econometric techniques were employed. The stationarity of the price time series was determined using the augmented Dickey–Fuller (ADF) unit root test. Structural breaks were inferred using the ADF test with a breaks test and a Bai–Perron multiple break test. The long-term relation/cointegration amongst the series was determined using the Johansen cointegration test (1988), with the previous breaks input as dummy variables. The direction of the causality was inferred using the Granger causality test (1969). The long-term and short-term causal relations were determined using the vector autoregression model (VAR) and the vector error correction model (VECM). The results showed a highly efficient and cointegrated market. The incidents of the trade war, as represented by tariffs and subsidies, had minor effects on the market efficacy, cointegration and price transmission. The arbitrage process of the studied market managed to get around the tariffs. In other words, there was no empirical evidence to support the claim that the law of one price (LOOP) did not hold.

**Keywords:** agricultural economics; econometrics; price transmission; soybean market; trade war; US–China trade war





## 1. Introduction

The trade war is an ongoing geopolitical conflict that originated with the Trump administration. This conflict involved US-initiated battles between the USA, China and US allies, causing repercussions throughout the entire world economy, the free trade system and the globalised world as we know it (Bown and Kolb 2020). Most sectors of the Chinese and US economies have been impacted during this war, and all have suffered the consequences of it to differing extents. Further, the magnitude of this phenomenon has impacted all world economies in some way, whether directly or indirectly, and it has generated high levels of volatility and uncertainty in the international and domestic markets, hampering trade and investment (Bown and Kolb 2020). On one hand, we have China, emerging as a rising economic power; whilst on the other hand, we have the USA, the first world economy in decline (Abdullahi and Phiri 2018). The Trump administration aimed to bend that decline and, as Trump's campaign quoted, "Make America great again". The US government intended to address this by boosting its economy by directly attacking the trade deficit, using extensive arsenals of importation tariffs, quotas, duties and threats to achieve a trade balance. The measures were imposed on China, as well as US allies. The Chinese government took these as direct attempts from the USA to eclipse the ascension of China as the first superpower and main world economy. They adopted different measures to respond to US policies.

The soybean market was in the spotlight during the first phase of the trade war since the Chinese government used this market and its market power as a leverage point to put pressure on the Trump administration. The international soybean market has been extensively studied for 50 years and empirical evidence suggests it is characterised as a highly efficient and cointegrated market, where the law of one price (LOOP) has been validated several times. The international soybean market has faced several market interventions from different players, such as "retentions" (export tariff) from Argentina (Margarido et al. 2007) or China's domestic price intervention by price support policies and flashing import tariffs to protect the domestic market (Arnade et al. 2017). In all these situations, the international and domestic prices have remained cointegrated, at least in the long term. The LOOP has prevailed, holding the price equalisation; however, the soybean market has never faced an exogenous event of this magnitude and extent. There is no empirical research confirming that US prices have dislocated from the international prices, away from the long-term equilibrium relationship, or that the LOOP was still valid or holding during the "soybean conflict". This constitutes a research gap into how this phenomenon induced volatility in this market and how structural breaks may compromise the significance of the cointegration vectors. This provides an investigative route into price transmission and cointegration research. There is vast empirical research regarding this market in terms of price transmission and cointegration; however, the research mainly focused on diagnosing the extent of global market cointegration, the loss of market, and market power in terms of price leadership. To provide an understanding of the subject, it is critical to introduce some basic concepts and definitions to the reader.

### 1.1. Critical Concept and Definitions

To understand the spatial (horizontal) price transmission or the co-movement of soybean prices in different market locations, the spatial arbitrage condition is a fundamental theoretical concept to comprehend (Listorti and Esposti 2012). This concept states that price differences in the same product between different markets cannot surpass the transaction cost, otherwise the arbitrage opportunity would be rapidly exploited. Derived from the *spatial arbitrage condition*, the law of one price (LOOP) states that discounting the transaction cost and when expressed in the same currency, two homogenous goods will have the same price. This concept was raised by Cournot (1838) and later Marshall (1890). The concepts of market efficiency and market integration are complementary to the LOOP. The first concept, namely, *market efficiency*, can be explained by the capability of markets to minimise the cost when the supply and demand are matched. Under the assumption of a competitive market with perfect information, the arbitrage process will help to reflect all the transaction costs. The concept of *market integration* focuses on the tradability of goods among a spatially separated market (Barrett and Li 2002).

### 1.2. Previous Empirical Research
#### 1.2.1. Law of One Price

There is vast empirical literature regarding the validity of the LOOP relating to the soybean market. Using the Johansen technique, Lima and Burnquist (1997) studied the validity of the previously mentioned law for the Brazilian, USA and German markets for the period 1985–1995 and the result showed the LOOP could not be rejected for those markets. Later, using a VECM, Margarido et al. (2001) studied the price transmission between the Brazilian market and the Rotterdam market, confirming the validity of the LOOP. Following this, Margarido et al. (2007) further extended the research by studying the price transmission of the soybean international market of Chicago, Rotterdam, Argentina and Brazil for the period between 1995 and 2003. Their research verified the validity of the LOOP for the soybean market by using cointegration and causality tests: the impulse response function (IRF), forecast error variance decomposition (FEVD) and the exogeneity test. de Sousa and Campos (2009) studied the price transmission of the Brazilian inter-regional market of soybean—specifically in the market involving Rio Grande do Sul, Parana

and Matogroso, and according to the authors, the LOOP could not be perfectly validated amongst all markets.

### 1.2.2. Price Transmission

Several researchers studied the different effects of soybean price transmission for international markets (horizontal price transmission among different countries) and different countries' domestic market supplies (vertical price transmission). The vast majority of the research focused on the elasticity, price formation, incomplete price transmission and asymmetry of price transmission of the main players within the market. Aguiar and Barros (1991) set the precedent for price transmission studies for the international market of soybean and the domestic market of Brazil. Their study exposed the causality relationship between both markets, where the international soybean market, represented by the Chicago Board of Trade (CBOT), leads international prices at least within the main markets (e.g., Rotterdam, Argentina and Brazil markets). This relationship was corroborated later by almost all researchers (Pino and Rocha 1994; Margarido and Sousa 1998; Margarido et al. 2007; Margarido 2012), who found that the United States, as represented by the CBOT, and the European Union, as represented by Rotterdam, led the international prices of this commodity. The authors identified Brazil and Argentina as "price takers". All the researchers found the international soybean market to be highly efficient and cointegrated (i.e., based on the market efficiency key performance indicators (KPIs), such as the speed of adjustment (error correction term) to shocks in the leading market.

### 1.3. Policy Implications in the International Market of Soybean: Price Transmission and Cointegration

To understand the real implications of the trade war on the international soybean market, it is necessary to understand the extent of this conflict and desegregate the main events that impacted or influenced the market from the secondary events that may have had minor importance. The soybean skirmish was composed of three main events. The first event occurred on 4 April 2018: the Chinese government imposed a 25% tariff on soybean. The second event occurred on 24 July 2018: the US government adopted subsidies for US farmers worth USD 27 billion. The third event occurred on 13 September 2019: the Chinese government lifted the tariff on imported US soybeans. Undeniably there were other minor factors in the soybean trade war, mostly in the form of threats, declarations and tweets from both sides, with the latter having the potential to increase the volatility of the future market and cause structural breaks. In order of magnitude, the aforementioned three main events were probably responsible for eventual price dislocation, loss of market efficiency and lack or partial temporal cointegration. The simplification of the "Soybean trade war" can go even further; it is possible to say that "essentially" this research is the study of a highly efficient market (soybean international market) in typically very volatile market conditions and under exogenous policy interventions of a tariff and a subsidy. Due to this, it is necessary to understand the theoretical effect of the policies in terms of price transmission and cointegration.

Empirical evidence showed that if two markets are linked by trade in a free market regime, the excess of demand or supply shocks will have an equal impact on prices for both markets (Mundlak and Larson 1992). The implementation of import tariffs on soybean into China would therefore be fully transmitted to the domestic prices of the commodity. An increase in tariff will generate a proportional increase in domestic price, which will be further transmitted to international prices before reaching an equilibrium point. However, this did not occur in the Chinese domestic market in April 2018, which can be explained by the price intervention policies that characterised this market (Zhao et al. 2010). This raises the first question: is the Chinese domestic market cointegrated with the international soybean market, or at least with the Chinese futures market (Dalian Futures)? There is empirical evidence to suggest that the degree of intervention in this market dislocated Chinese domestic prices from the international price, at least in the short term (Arnade et al. 2017);

therefore, the Chinese price should not be affected in the short term. The rest of the international prices were highly cointegrated and the LOOP was validated; therefore, a rise in international prices from April 2018 would have been expected. However, this did not happen until July 2018 when the Chinese tariff was made effective. When the increase in tariff was prohibitively high, this could partially dislocate the domestic and international market prices and stimulate both to move independently of each other, preventing price convergence (Gardner 1975; Mundlak and Larson 1992; Quiróz and Soto 1995; Lima and Burnquist 1997; Baffes and Ajwad 2001; Abdulai 2007). This last scenario is what may have happened in July 2018 when domestic US prices were dislocated from international prices. It is well-known that policy interventions can affect the price transmission dynamics and spatial arbitrage, especially in cross-country price transmission. Policies such as export subsidies, variable levies, tariff rates and quotas, non-tariff barriers and prohibitive tariffs may prevent price convergence. Fixed tariffs are expected to affect the price spread by acting as fixed or proportional transaction costs (Listorti and Esposti 2012).

The trade war scenario is more complex than a simple soybean imports tariff. A standard import tariff acts as a fixed transaction cost that allows price transmission to occur in the international soybean market (Rapsomanikis et al. 2013). However, taking the trade war scenario, the tariff was exclusively assigned to the USA, meaning only the USA export market faced a 25% tariff on their exported soybeans to China. Considering the volume of USA soybean production that is exported to China (two-thirds of the USA's exported soybeans), this tariff acted as a fixed transaction cost, forcing Chinese importers to switch to alternative suppliers. When the tariff was announced on 4 April, all markets were falling, despite the fact that the US spot and US futures markets were the only ones affected by the tariff. The future market plays the role of future price expectations and is affected by future tariffs. It might seem sensible to think that other markets could not be affected by future pessimistic market expectations; however, the future market leads spot prices, which was demonstrated by previous empirical research; the Chicago future prices lead the Brazilian and Argentinian prices (Margarido et al. 2007).

Chicago is the main international market leading the prices. Since the tariff exclusively affected the USA, other countries have not been targeted, hence their own market has not suffered directly from a transaction cost. Although Chicago leads the international prices, and almost all markets are integrated (follow a long-term relationship), by the end of June 2018, the market showed the first sign of lack of convergence. By mid-July, the international prices were seen to be dislocated from the USA soybean price (Figure 1). The United States was undergoing a sustained fall, whereas the other international prices were actively recovering. The USA's soybean exports diverged from China's market (the largest importer market in the world) due to the high transaction cost. The USA's impossibility of selling its surplus production to the main market drove its soybean prices to drastically fall in US markets. Simultaneously, this generated a shortage in supply in the Chinese feed industry, where the Chinese domestic market had no capability to fill the gap left by US soybeans. This made Chinese importers switch to the Brazilian market, which is a major soybean producer in the world and the only market capable of filling the void left by the United States (as their main supplier). When the Chinese importers redirected demand to Brazil, this generated a price increase in the Paranaguá market (Brazil). In turn, this generated a price displacement in the international market; whilst the US price was actively falling, the Brazilian price was peaking, generating a price dislocation at the end of June 2018. Interestingly, Argentina's price increased too, suggesting that it was following the Brazilian price. As previously mentioned, according to Margarido et al. (2007), Argentina's prices follow the Chicago price; therefore, theoretically, the Argentinian price should have followed the US fall. However, the price dislocation altered the price leadership, and it seems, at least for a short period, that Brazil led the international price.

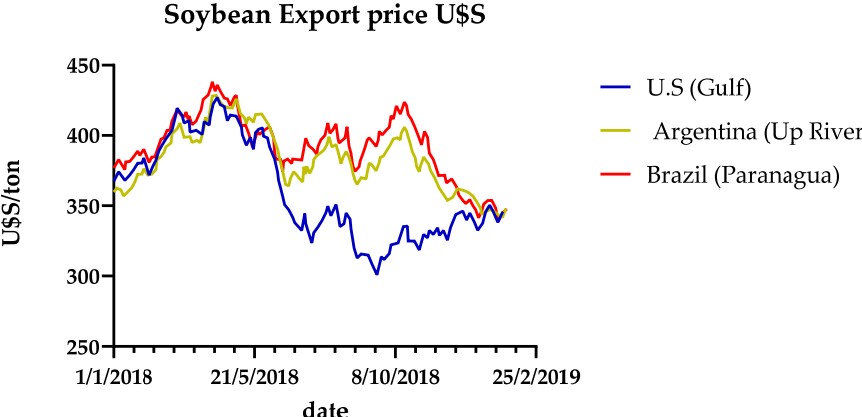

**Figure 1.** Soybean price time series.

The price dislocation obeyed the mathematical principle of lack of cointegration vectors; this dislocation process means that the prices do not follow a long-term equilibrium. The US prices were dislocated from the international prices, at least in the short term. If the divergence is only in the short term, this means that the price may still be cointegrated. By mid-December 2018, the gap between different international prices closed. This showed that for 4.5 months, the international prices may not have been cointegrated with the USA prices. Over time, this gap was closed by the ability of the market to arbitrage around the tariff. The fall in US prices increased the competitiveness of US soybeans, allowing the production to be allocated to other destinations (Argentina, Iran and Europe); however, this was not enough to cover the loss of the Chinese market (two-thirds of USA soybean exports and two-thirds of the global demand). Europe has historically been supplied by Brazilian soybeans, but following the Brazilian price increase, Europe started to substitute Brazilian soybean and rapeseed for the low-cost US soybean (Setser 2019). In addition, the market arbitrage opportunity was taken advantage of by Argentina, using the high crushing capacity. Argentina imported the US low-cost soybean, crushing it to be re-exported as soymeal. All the evidence supports the idea that the markets exploited the arbitrage opportunities. The famous "invisible hand" regulated the market (Smith 1723), in other words, an example of the applicability of the LOOP and a highly efficient market.

By 24 July 2018, the second main event occurred. The Trump administration an-nounced a compensation subsidy for farmers affected by the US soybean import tariffs. In other words, Trump implemented an export subsidy because the majority of US soybean was exported and, as a result, the US price increased in the following days, going against the economic theory and empirical evidence. Previous researchers demonstrated that when an export subsidy is implemented, the price tends to decrease (Kerkelä et al. 2005); however, this economic phenomenon did not occur in the short term, as there was a lag period where the economic agents adapted to the subsidy. Due to this, it is possible to infer that this notorious increase in price can be associated with the US futures market speculation process, or most probably the arbitrage process was already equalising international prices. This second theory is more likely: the swift demand from other countries was already leveraging prices in a display of market cointegration, cross-country price transmission and efficiency. The third event of the trade war was the tariff lift, which is believed to have had an almost null effect because, in that period, the international prices had already converged, the world supply and demand were already reorganised and the import tariff was acting as a transaction cost for the Chinese importers (although, Chinese importers were importing soybeans from other markets to avoid these). Despite this, it is believed that the tariff lift increased the market efficiency and economic welfare: Brazilian and US soybeans have different harvesting seasons, allowing importers to have a constant flow of soybean throughout the year with less price volatility and price seasonality.

In summary, this price convergence is a historical example of market efficiency and efficient price transmission that was reviewed in this investigation. By February 2019, all prices appeared to return to a long-term equilibrium, obeying the "Law of One price". The market reorganised around the tariff. The complexity of this event opened the research question: to what extent was the international market affected by the trade war? The evidence seems to highlight that in a fully integrated market, market power can be easily overcome by avoiding asymmetrical price transmission or price dislocation. What happened during this historical event was the perfect example. The tariff counterpart, namely, the subsidy, which seemed to be useless, probably contributed in the long term to generating a US soybean surplus to flood the international market and drop the international prices, benefiting only US farmers and international soybean final consumers to the detriment of all international soybean producers and US taxpayers.

Since this geopolitical conflict is relatively new, there is no empirical research that verifies that the trade war induced breaks in the markets, or that the market was partly dislocated in the short term and "The Law of One Price" prevailed in the end. Looking at the time price series, we can draw those conclusions without any complex mathematical and economical models; however, in science, things cannot be taken for granted and this is how our research began, perhaps attempting to verify the obvious, but the obvious cannot always be assumed.

This research attempted to analyse the impact of the trade war on the price system of the international market of soybean by using the latest econometric models and statistical analysis. To achieve this, it was first necessary to demonstrate market cointegration and understand the market causal relation and market efficiency so that we could later analyse how the exogenous shocks correlated with the trade war events affected the market price system.

## 2. Methodology

This research was fundamentally based on econometrics models based on time series. The secondary data came from several sources, such as the international monetary fund (IMF), Chicago Board of Trade (CBOT), CEPEA Brazil (Centro de Estudos Avançados em Economia Aplicada), Dalian Commodity exchange (China), Wind Economic Database (China) and Bolsa de Rosario (Argentina). All the time series were converted to monthly data expressed in the same currency and volume (USD/tonne), deflated and transformed into a natural logarithm. These adjustments were required to remove sources of variation, simplify patterns in the time series and increase the accuracy of the models. The evaluated period was from September 2009 to May 2019.

The econometric procedure followed a logical sequence. First, it was necessary to determine the order of integration of the time series; to do this, this research used the augmented Dickey and Fuller (1979) (ADF) stationarity test. Following this procedure, it was necessary to find structural breaks in the time series using the ADF-modified version of the test with breaks and Bai–Perron multiple breaks detection. After the breaks detection, it was necessary to infer cointegration between the time series using the Johnsen cointegration test, and if possible, to infer this mathematical property between the time series. However, cointegration and correlation do not imply causality; therefore, a Granger causality test was required to infer this property. The econometrics models selected helped to identify the effects the geopolitical situation caused, for instance, the cointegration test showed the cointegration amongst the markets. The lack of cointegration means that the markets were not integrated, i.e., they no longer followed a long-term equilibrium relationship. This can be implicitly associated with the trade war if the trade war spillover effect was significant in causing a partial or permanent dislocation of the markets. Previous researchers found and verified the cointegration amongst the soybean global markets. Margarido (2012) systematically investigated this mathematical property across different periods and geopolitical situations, reaching the same conclusion. Therefore, the cointegration test result can be a rigorous indicator of market integration. The absence of cointegration can be correlated

with market dislocation or the absence of a free market regime, which can be caused by external shocks, such as the trade war. The unit root test (with breaks) is a stationarity test, and although it has low statistical power, it can be a useful indicator of exogenous shocks in the market. These exogenous shocks can introduce unit roots in the time series transforming the series into a non-stationary one. The process that then follows is the finding of the breaks (unit root with breaks test), which are then input as dummy variables into the cointegration test. When the structural breaks are integrated into the cointegration test as dummy variables, and if those breaks are statistically significant, the cointegration vector can stand out, highlighting cointegration between the series. The time series break may be later associated with the trade war events or other exogenous events. This process validates the importance of detecting structural breaks in time series, as this may disrupt the market by introducing unit roots and dislocating the time series (i.e., non-cointegration). According to Engle and Granger (1987), the cointegration concept has relevant implications, which are proposed in the Granger representation theorem. This states that if two trending variables integrated with order I(1) are cointegrated, the relationship between the variables can describe an error correction model (VECM) (Listorti and Esposti 2012).

An error correction model (ECM) can be critical to assessing the market efficiency loss, and this phenomenon can later be associated with the trade war spillover. Using the speed of adjustment coefficient of the error correction term represented by the Greek letter $\lambda$ (lambda), which is commonly known as the error correction term, it is possible to infer the market efficiency. This coefficient indicates the speed at which a market returns to the long-term equilibrium and will solely depend on the proximity of this coefficient to one. This proximity to one can be used to assess the extent to which distortions, transaction costs and policies delay the price adjustment to the long-term equilibrium. The study undertaken by Sharma (2002) is an example of assessing market policy implications through market integration, as estimated using error correction models. According to the author, in countries where the government intervenes through several policies, the error correction coefficients clearly indicated a slow speed of adjustment to the long-run equilibrium (lambda: $(-0.01)$–$(-0.07)$ or $1-7\%$ per period).

From the construction of a vector error correction model, it is possible to extract short- and long-term regression coefficients. Using short-term equation regressor coefficients, it is possible to diagnose the short-term causality (in the Granger sense) and the short-run effects. The long-term causality can be inferred by the statistical significance of the long-term equation regressors. The Granger causality test is a useful methodology to map the causality of interconnected markets. It is possible to test the markets in pairs to infer the direction of the causality and recreate the market power dynamic. Previous researchers already mapped this causality for the period before and after the trade war, from which we can denote the change in causality. These might be correlated with the change in market power caused by the trade war. The VECM can provide a stylised image of the relationships between and within the markets. In addition, the error correction representation can be used as a framework for testing nonlinear adjustment to the long-term equilibrium (asymmetric price transmission) (Rapsomanikis et al. 2013).

This research was designed to use several econometrics models and tests at different stages of the study. In the first stage of the investigation, the author only used some of the econometric techniques mainly for price transmission research. Only the methodology used in this study will be explained. The mathematical theory behind the models is extensive and could easily drive the attention of the reader to technicalities and formalities; thus, the author assumes that readers have a technical understanding of econometrics and market dynamics. Therefore, the author has chosen to omit the underpinning mathematical–statistical theory and the explanation of the econometric models while, however, keeping the essential information. In contrast, it can be suggested that the main weakness of this study is the methodology since the econometric procedure might lack statistical power or the regression model might not prove causation.

*2.1. Hypothesis*

The following sections introduce the hypotheses used within this research.

2.1.1. Market Cointegration and Structural Breaks Paradigm

If the occurrence of the trade war has drastically affected the market prices, it should be possible to assume that this phenomenon has induced a structural break in the time series. As such, the research question is as follows: did the outbreak of the trade war induce a structural break and affect the long-term relations between the markets? The null hypotheses were as follows:

- N0.1. The markets were not dislocated, i.e., there was cointegration between the markets.
- N0.2. The trade war did not introduce a break in the soybean market time series; therefore, the time series were stationary at first difference.

2.1.2. The Market Efficiency Affectation

Complete price transmission and a high speed of adjustment clearly reflect the market efficiency. Furthermore, the validity of the law of one price is a consistent indicator of an efficient market. The research question concerns whether the efficiency of the market was affected. This research question generated several null hypotheses:

- N0.1. The price transmission within the market was not affected.
- N0.2. The law of one price held in the USA–China trade war.

## 3. Results

This chapter introduces the results of the econometric tests in the same logical order they were performed. They are summarised in table format to improve readability and to aid comprehension.

*3.1. Unit Root Test*

The first test performed was the unit root test. This test is critical for understanding the suitability of the data for different econometric procedures and econometric pathways. The ADF (unit root test) results for the seven time–price series (at level) showed that it was not stationary (Table 1). For all the series, the null hypotheses could not be rejected because the probability levels were higher than the critical level in all cases, meaning a high probability of committing a type I error (rejecting the null hypothesis when it is true). Therefore, it was necessary to transform the series to the first difference and rerun the test. After the time series were differentiated in the first difference, the ADF showed that they were stationary (Table 2). In conclusion, the t-stat was less than the test critical value in all cases at the 5% significance level. Furthermore, in all cases, the probability was less than 0.05 (5%); therefore, the null hypotheses were rejected and the alternative hypotheses were accepted. The test showed that all series were integrated with order 1-I(1).

**Table 1.** Unit root test level.

| Price Series | Test Critical Value 5% | | ADF Test Statistic | Prob. | Null Hypothesis |
|---|---|---|---|---|---|
| LNCHIGF | −2.88 | < | −2.09 | 0.25 | Accepted |
| LNCHINASP | −2.88 | < | −1.47 | 0.55 | Accepted |
| LNDALIF | −2.88 | < | −1.83 | 0.36 | Accepted |
| LNPARANAGUASP | −2.88 | < | −2.25 | 0.19 | Accepted |
| LNROSFT | −2.88 | < | −2.55 | 0.11 | Accepted |
| LNROSSP | −2.88 | < | −2.37 | 0.15 | Accepted |
| LNROTTERDAM | −2.88 | < | −1.41 | 0.57 | Accepted |

**Table 2.** Unit root test first difference.

| Price Series | Test Critical Value 5% | | ADF Test Statistic | Prob. | Null Hypothesis |
|---|---|---|---|---|---|
| LNCHIGF | −2.88 | > | −8.13 | 0.00 | Rejected |
| LNCHINASP | −2.88 | > | −8.06 | 0.00 | Rejected |
| LNDALIF | −2.88 | > | −10.26 | 0.00 | Rejected |
| LNPARANAGUASP | −2.88 | > | −7.42 | 0.00 | Rejected |
| LNROSFT | −2.88 | > | −7.44 | 0.00 | Rejected |
| LNROSSP | −2.88 | > | −7.99 | 0.00 | Rejected |
| LNROTTERDAM | −2.88 | > | −9.17 | 0.00 | Rejected |

*3.2. Structural Breaks Test*

Structural breaks in a time series can introduce a false unit root, leading to a lack of cointegration vectors; due to this, it is necessary to examine the occurrence of structural breaks. Moreover, the importance of finding structural breaks for later correlation with the events of the trade war is crucial for assessing the importance of the trade war on the international markets. The technique that was used to infer the structural breaks in the series was the Bai–Perron multiple break test (Table 3) (using the Schwarz and LWZ criteria) and the ADF test (unit root test with breaks) (Table 4). The test results for all the time series for the entire period showed a total of 49 breaks, with several being repeated, depending on the test, test criteria and time series. Only 17 were unique breaks in the collection of time series, of which all the breaks could be correlated with different exogenous shocks in the international soybean market. Different periods showed a higher frequency of market breaks; the higher frequencies were found around the last quarter of 2010, across 2012 and during 2014. The breaks were associated with climatic disasters, such as drought, and stock deficits or speculation processes. Only a single structural break was correlated with the trade war event in the Dalian Futures market in February 2018. Despite the fact that the trade war effectively started in March 2018 with Trump's tariff announcements, traders may have reacted to market rumours and/or inside information acting in anticipation and provoking a structural break in the market. However, the affectation in terms of external shocks that generated breaks in the market can be considered minimal for the studied period; this decreased the importance of the trade war in relative terms when considering other events of the studied period.

**Table 3.** Bai–Perron test multiple breakpoint test.

| Price Series | Breaks | of Coefs. | Sum of Sq. Resids | Log-L | Schwarz Criterion | LWZ Criterion | Schwarz Breaks | LWZ Breaks |
|---|---|---|---|---|---|---|---|---|
| LNCHIGF | 2 | 5 | 0.728 | 140.92 | −4.93 | −4.75 | 2010M10 | 2010M11 |
| | 3 | 7 | 0.630 | 149.84 | −5.00 | −4.75 | 2012M04 | 2014M09 |
| | | | | | | | 2014M09 | |
| LNCHINASP | 3 | 7 | 0.251 | 206.19 | −5.91 | −5.67 | 2010M11 | 2010M11 |
| | 4 | 9 | 0.219 | 214.56 | −5.97 | −5.65 | 2012M08 | 2012M07 |
| | 5 | 11 | 0.198 | 221.00 | −6.00 | −5.61 | 2014M04 | 2015M10 |
| | | | | | | | 2015M10 | |
| | | | | | | | 2017M10 | |
| LNDALIF | 3 | 7 | 0.429 | 173.41 | −5.38 | −5.13 | 2010M09 | 2010M09 |
| | 4 | 9 | 0.385 | 180.11 | −5.41 | −5.09 | 2012M03 | 2012M03 |
| | | | | | | | 2015M09 | 2015M10 |
| | | | | | | | 2017M10 | |
| LNPARANAGUASP | 3 | 7 | 0.806 | 134.63 | −4.75 | −4.50 | 2010M09 | 2010M09 |
| | | | | | | | 2012M04 | 2012M04 |
| | | | | | | | 2014M09 | 2014M09 |
| LNROSFT | 3 | 7 | 0.579 | 154.96 | −5.08 | −4.83 | 2010M10 | 2010M10 |
| | | | | | | | 2014M09 | 2014M09 |
| | | | | | | | 2016M05 | 2016M05 |

**Table 3.** *Cont.*

| Price Series | Breaks | of Coefs. | Sum of Sq. Resids | Log-L | Schwarz Criterion | LWZ Criterion | Schwarz Breaks | LWZ Breaks |
|---|---|---|---|---|---|---|---|---|
| LNROSSP | 2 | 5 | 1.00 | 120.91 | −4.60 | −4.37 | 2010M10 | 2010M10 |
| | 3 | 7 | 0.916 | 126.75 | −4.62 | −4.28 | 2012M04 2014M07 | 2014M08 |
| LNROTTERDAM | 2 | 5 | 0.639 | 148.94 | −5.064 | −4.88 | 2010M10 2014M08 | 2010M10 2014M08 |

Minimum information criterion values are highlighted in yellow.

**Table 4.** Unit root test with breaks.

| Price Series | Test Critical Value 5% | | ADF Test Statistic | Prob. | Break Date |
|---|---|---|---|---|---|
| LNPARANAGUASP | −4.44 | > | −8.15 | <0.01 | 2012M07 |
| LNROTTERDAM | −4.44 | > | −9.74 | <0.01 | 2012M07 |
| LNCHIGF | −4.44 | > | −8.64 | <0.01 | 2014M09 |
| LNROSFT | −4.44 | > | −8.050 | <0.01 | 2016M05 |
| LNROSSP | −4.44 | > | −8.41 | <0.01 | 2016M05 |
| LNCHINASP | −4.44 | > | −8.83 | <0.01 | 2016M09 |
| LNDALIF | −4.44 | > | −11.50 | <0.01 | 2018M02 |

*3.3. Cointegration Test*

The cointegration determination is critical to assess the impact of the trade war; however, it is necessary to first check the assumption that all prices were cointegrated, thereby moving together and following a long-term relationship. This last mathematical property is fundamental to understanding whether the different events of the trade war provoked market dislocation and non-cointegration in the process. Using the Johansen cointegration test, which was performed in pairs for all the time series, the cointegration was checked, forming the cointegration matrix in Table 5. The primary conclusion was that not all the time series were cointegrated. This lack of cointegration could be explained by government intervention, such as in the cases of China and Argentina. These market interventions may cause price dislocation and loss of market efficiency. Market interventions could have caused structural breaks in the time series, and by using these breaks as dummy variables in the Johansen cointegration model, it was possible to find cointegration. The next step of the research was to re-run the cointegration analysis using the previous breaks as dummy variables. Using this technique, it was possible to find full cointegration between all the time series (Appendix A Table A1).

**Table 5.** Johansen cointegration test: unrestricted cointegration rank test.

| Unrestricted Cointegration Rank Test (Trace) | | | | | | | |
|---|---|---|---|---|---|---|---|
| | LNCHIGF | LNCHINASP | LNDALIF | PARANAGUASP | LNROSFT | LNROSSP | ROTTERDAM |
| LNCHIGF | X | 1 | 1 | 1 | 0 | 1 | 1 |
| LNCHINASP | 1 | X | 1 | 0 | 0 | 1 | 2 |
| LNDALIF | 1 | 1 | X | 1 | 0 | 1 | 1 |
| LNPARANAGUASP | 1 | 0 | 1 | X | 0 | 1 | 1 |
| LNROSFT | 0 | 0 | 0 | 0 | X | 1 | 0 |
| LNROSSP | 1 | 1 | 1 | 1 | 1 | X | 1 |
| LNROTTERDAM | 1 | 1 | 1 | 1 | 0 | 1 | X |

The numbers in the table represent the number of cointegration equations at the 0.05 level.

After re-running the Johansen cointegration test with the previously inferred structural breaks, we were able to find full cointegration between all the series (Table 6). However, the break associated with the trade war was not able to act as a dummy variable that would help to infer cointegration vectors from the Dalian Futures. This fact relativised the impact of the trade war as a disruptive factor that dislocated the international market of

soybean. In other words, there were other exogenous events that had a stronger impact and dislocated the market in the studied period.

**Table 6.** Johansen cointegration test: unrestricted cointegration rank test adjusted using breaks.

| Unrestricted Cointegration Rank Test (Trace) | | | | | | | |
|---|---|---|---|---|---|---|---|
| | **LNCHIGF** | **LNCHINASP** | **LNDALIF** | **LNPARANAGUASP** | **LNROSFT** | **LNROSSP** | **LNROTTERDAM** |
| LNCHIGF | X | 1 | 1 | 1 | 1 | 1 | 1 |
| LNCHINASP | 1 | X | 1 | 1 | 1 | 1 | 2 |
| LNDALIF | 1 | 1 | X | 1 | 1 | 2 | 1 |
| LNPARANAGUASP | 1 | 1 | 1 | X | 1 | 1 | 1 |
| LNROSFT | 1 | 1 | 1 | 1 | X | 1 | 1 |
| LNROSSP | 1 | 1 | 2 | 1 | 1 | X | 1 |
| LNROTTERDAM | 1 | 2 | 1 | 1 | 1 | 1 | X |

The numbers in the table represent the number of cointegration equations at the 0.05 level.

### 3.4. Granger Causality Test

The Granger causality test was critical in this research to understand the dynamics of the market prices in terms of causality; this test allowed us to create a causal map of the market. The optimal lag selection was inferred using the Swartz information criterion and, according to this criterion, the optimal lag selection was one lag. The Granger causality matrix (Table 7) summarises the causality interactions for the optimal lag selection (one lag). The causality dynamic seemed to agree with the previous research. It can be clearly appreciated that the Chicago and Rotterdam prices led the markets without any trace of causality between them. It is noticeable that the Brazilian market of Paranaguá seemed to affect several international prices, such as the Rosario Spot, as well as the Chinese, Dalian Futures and China Spot markets. There is no previous empirical research suggesting the rise of the Brazilian market as a price leader. This creates an open question: when and how has the Brazilian market risen as a leading market? The sensible answer is that the increase in the Brazilian domestic market and crushing industry, on top of being the main soybean exporter of the world, progressively made Brazil a leading market. However, to what extent the trade war effect had leverage on the Brazilian market as an international leader is one undetermined factor.

**Table 7.** Granger causality test (matrix).

| Granger Causality Test with One Lag | | | | | | | |
|---|---|---|---|---|---|---|---|
| | **LNCHIGF** | **LNCHINASP** | **LNDALIF** | **LNPARANAGUASP** | **LNROSFT** | **LNROSSP** | **LNROTTERDAM** |
| LNCHIGF | X | ↑ | ↑ | ↑ | ∂ | ↑ | ∂ |
| LNCHINASP | ← | X | ← | ← | ← | ← | ← |
| LNDALIF | ← | ↑ | X | ← | ∂ | ← | ← |
| LNPARANAGUASP | ← | ↑ | ↑ | X | ∂ | ↑ | ← |
| LNROSFT | ∂ | ↑ | ∂ | ∂ | X | ↑ | ∂ |
| LNROSSP | ← | ↑ | ↑ | ← | ∂ | X | ← |
| LNROTTERDAM | ∂ | ↑ | ↑ | ↑ | ∂ | ↑ | X |

The direction of the arrows (←, ↑) represents the direction of the causality. ∂ represents no causality affectation.

Knowing that almost all series were cointegrated, and after the empirical results demonstrated the causal relationship between the different market time series, it was possible to develop a vector error correction model to capture the cointegration, the causal relation and price transmission.

### 3.5. Vector Error Correction Model

Another perspective to consider is the influence the trade war had on market efficiency. The vector error correction model is an important tool to assess the market efficiency through the price transmission and market power through implied causality on the short- and long-term equations. The result of the VEC model is an equation system, where the model incorporates the short-term and long-term relationships of the series. This methodology was applied in pairs to map individual and paired interactions. Following

the same logic, global models were created to understand the compound dynamics of each time series. Several diagnostic tests were applied to avoid spurious regressions and assure the reliability of the results. Residual serial correlations were checked using the LM autocorrelation test and normality using the Cholesky (Lutkepohl) orthogonalisation test. The least squares regression in conjunction with the Wald test was applied to find the significance of the regressors.

### 3.5.1. VECM China Spot Market Pair Model

Several VEC models were built that embedded the Granger causality (g-causality) relationship and the cointegration between the time series (Table 8). As previously mentioned, in terms of the Granger causality, the China Spot market did not affect any of the studied markets. In contrast, the China Spot market was affected by almost all the investigated markets. This finding was coherent with the level of government intervention in this market. Despite the Chinese soybean importation tariffs being relatively low, the Chinese government used several other mechanisms to intervene to influence the prices., The Chinese policymakers probably used international prices to fix the domestic prices to some degree. This may be the reason behind the China Spot market showing itself as a weak market that tended to follow the others. The degree of intervention can clearly reflect the loss of market efficiency, which can be captured by the speed of adjustment of the error correction term (VECM). For all the different models created, the speed of adjustment was extremely low, varying between 1.27–5.46% per period, indicating poor market efficiency. The China Spot price seemed to adjust faster to the Chicago prices (5.46%), meaning that a shock in prices in Chicago resulted in China's domestic soybean price returning to the long-term equilibrium in 18 months, denoting a poor market efficiency and incomplete price transmission. For all the different model combinations, the short-term regressors were not significant; only the Chicago price at an alpha of 10% was significant. This suggested that in the short-term, China tended to dislocate from the international prices, but to some degree, the Chicago price influenced the Chinese policymaker in the short term. In the long-term, however, the Chinese price tended to follow and adjust based on international prices led by Chicago. The minimum price program was created by the Chinese government and was designed to adapt to market liberalisation. Other measures, such as "target price" and "deficit payment", were measures created to compensate Chinese farmers for international price fluctuation (Jamet and Chaumet 2016). All these measures shielded the market from international price fluctuations and interrupted efficient price transmission. The VEC models showed that Chicago prices could only explain 7.51% of the China Spot prices and Rotterdam and Rosario could explain 6.48% and 5.54% of the price increases, respectively. Dalian Futures could be associated with decreases in price in the China Spot market, explaining 1.73% of the price decreases.

Equation (1): VECM corrected using coefficient significance

$$\Delta \text{LNCHINASP} = -0.055(\text{LNCHINASP}_{t-1} - 0.95\text{LNCHIGF}_{t-1} - 0.73) +$$
$$0.185\Delta \text{LNCHINASP}_{t-1} + 0.075\Delta \text{LNCHIGF}_{t-1} \tag{1}$$
$$\lambda = -0.0546$$

Equation (2): VECM corrected using coefficient significance

$$\Delta \text{LNCHINASP} = -0.103(\text{LNCHINASP}_{t-1} - 0.97\text{LNDALIF}_{t-1} - 0.21) + 0.29\Delta \text{LNCHINASP}_{t-1}$$
$$\leftarrow \textit{No short-term regressors from Dalian Futures} \tag{2}$$
$$\lambda = -0.103251$$

Equation (3): VECM corrected using coefficient significance

$$\Delta \text{LNCHINASP} = -0.047(\text{LNCHINASP}_{t-1} - 1.033\text{LNPARANAGUASP}_{t-1} - 0.173) + 0.21\Delta \text{LNCHINASP}_{t-1}$$
$$\leftarrow \textit{No short-term regressors from Paranguas} \tag{3}$$
$$\lambda = -0.047338$$

Equation (4): VECM corrected using coefficient significance

$$\Delta LNCHINASP = -0.013(LNCHINASP_{t-1} - 2.43LNROSFT_{t-1} + 7.26) + 0.24\Delta LNCHINASP_{t-1}$$
$$\leftarrow \textit{No short-term regressors from Rosario Futures}$$
$$\lambda = -0.012727$$
(4)

Equation (5): VECM corrected using coefficient significance

$$\Delta LNCHINASP = -0.032(LNCHINASP_{t-1} - 1.266LNROSSP_{t-1} + 0.715) + 0.256\Delta LNCHINASP_{t-1}$$
$$\leftarrow \textit{No short-term regressors from Rosario Spot}$$
$$\lambda = -0.031843$$
(5)

**Table 8.** Vector error correction pairs model for China Spot.

| | | | | | | | | Heteroskedasticity Tests | |
|---|---|---|---|---|---|---|---|---|---|
| | Lag | D(LNCHINASP) | < | D(LNCHIGF) | Elasticity | Correlation LM Tests | Normality Test | Without Cross Term | With Cross Term |
| CointEq1 | 1 | −5.46% | | 7.51% | 1:0.95 | 0.9724 | 0.00000 | 0.9270 | 0.9662 |
| | | D(LNCHINASP) | < | D(DALIF) | | | | | |
| CointEq1 | 1 | −10.33% | | −1.73% | 1:0.97 | 0.5821 | 0.00000 | 0.2897 | 0.2932 |
| | | D(LNCHINASP) | < | D(PARANAGUASP) | | | | | |
| CointEq1 | 1 | −4.73% | | 5.04% | 1:1.03 | 0.9341 | 0.00000 | 0.6773 | 0.6915 |
| | | D(LNCHINASP) | < | D(ROSFT) | | | | | |
| CointEq1 | 1 | −1.27% | | 6.48% | 1:2.43 | 0.9864 | 0.00000 | 0.4850 | 0.2489 |
| | | D(LNCHINASP) | < | D(ROSSP) | | | | | |
| CointEq1 | 1 | −3.18% | | 1.45% | 1:1.26 | 0.9048 | 0.00000 | 0.3379 | 0.6308 |
| | | D(LNCHINASP) | < | D(ROTTERDAM) | | | | | |
| CointEq1 | 1 | −5.02% | | 5.54% | 1:1.12 | 0.8665 | 0.00000 | 0.8224 | 0.8361 |

The creation of a global model for the China Spot market resulted in a spurious regression (Table 9). All the coefficients failed to be significant, and the residuals were not normally distributed. These can be considered expected and intuitive results, as the slow speed of adjustment and the low determination coefficient from the model pairs were in concordance with a high degree of government intervention. This market seemed to be dislocated from the international prices in the short term and, to some degree, was affected by the Chicago prices.

**Table 9.** Vector error correction global model for China Spot.

| CointEq1 | Lag | ΔLNCHINASP | < | ΔLNCHIGF | ΔDALIF | ΔPARANAGUASP | ΔROSFT | ΔROSSP | ΔROTTERDAM |
|---|---|---|---|---|---|---|---|---|---|
| λ | 1 | −1.40% | | 15.64% | 0.10% | 0.50% | −4.81% | −3.74% | 3.56% |
| | | | | Correlation LM | Normality Test | Without Cross Term | With Cross Term | | |
| | | | | 0.32740 | 0.00000 | 0.13050 | 0.05870 | | |

### 3.5.2. VECM Dalian Futures

The error correction model for Dalian Futures for the different pair combinations showed an overall higher speed of adjustment and market efficiency than the China Spot market (Table 10). The lambda coefficient from the error correction term fluctuated between 13.8% and 7.5%. The Chicago and Rotterdam models showed higher adjustment speeds, with 13.7% and 13.5%, respectively. Chicago could be associated with an increase in the price of 4.58% and Rotterdam with an increase of 6.45%. This higher market efficiency could be associated with a lower degree of market intervention; however, in contrast with previous empirical evidence, this speed of adjustment was relatively slow. The short-term equation for all VEC pair models did not show any short-run regressors, implying a meaningless, short-term, causal relationship with the entire market. The creation of a

global model incorporating all the cointegrated series resulted in a spurious regression, even at an alpha of 10%; therefore, it was not included in this investigation.

**Table 10.** Vector error correction pairs model for Dalian Futures.

| | | | | | | | | Heteroskedasticity Tests | |
|---|---|---|---|---|---|---|---|---|---|
| | Lag | ΔLNDALIF | < | ΔLNCHIGF | Elasticity | Correlation LM Tests | Normality Test | Without Cross Term | With Cross Term |
| CointEq1 | 1 | −13.76% | | −4.58% | 1:0.79 | 0.3891 | 0.00000 | 0.4118 | 0.3303 |
| | lag | ΔLNDALIF | < | ΔPARANAGUAS | | | | | |
| CointEq1 | 1 | −10.60% | | −3.36% | 1:0.89 | 0.2002 | 0.00000 | 0.2837 | 0.4106 |
| | lag | ΔLNDALIF | < | ΔROSSP | | | | | |
| CointEq1 | 1 | −7.53% | | −3.07% | 1:1.06 | 0.3251 | 0.00000 | 0.3923 | 0.5137 |
| | lag | ΔLNDALIF | < | ΔROTTERDAM | | | | | |
| CointEq1 | 1 | −13.52% | | −6.45% | 1:0.89 | 0.2351 | 0.00000 | 0.5950 | 0.8554 |

Equation (6): VECM corrected using coefficient significance

$$\Delta LNDALIF = -0.138(LNDALIF_{t-1} - 0.793LNCHIGF_{t-1} - 1.672)$$
$$\leftarrow \textit{No short-term regressors}$$
$$\lambda = -0.138$$
(6)

Equation (7): VECM corrected using coefficient significance

$$\Delta LNDALIF = -0.106(LNDALIF_{t-1} - 0.888LNPARANAGUASP_{t-1} - 1.051)$$
$$\leftarrow \textit{No short-term regressors}$$
$$\lambda = -0.106$$
(7)

Equation (8): VECM corrected using coefficient significance

$$\Delta LNDALIF = -0.135(LNDALIF_{t-1} - 0.868LNUSROTTERDAMCIF_{t-1} - 1.122)$$
$$\leftarrow \textit{No short-term regressors}$$
$$\lambda = -0.135$$
(8)

Equation (9): VECM corrected using coefficient significance

$$\Delta LNDALIF = -0.075(LNDALIF_{t-1} - 1.065LNROSSP_{t-1} - 0.421)$$
$$\leftarrow \textit{No short-term regressors}$$
$$\lambda = -0.075$$
(9)

### 3.5.3. VECM Paranaguá

The Paranaguá overall VEC model showed a high adjustment speed in the error correction term, with 27.32% for Rotterdam and 26.80% for Chicago (Table 11). Rotterdam appeared to lead the prices for this market, and this was in concordance with previous empirical research. (Margarido et al. 2007) explained that the Rotterdam price led the Brazilian market since the price formation was on the demand side. In other words, if a shock in the Chicago or Rotterdam market occurred, Paranaguá would return to the long-term relationship in around 3 months and two weeks. However, all model pairs failed to present short-term significant regressors; thus, there was no evidence of short-term causality between Rotterdam and Paranaguá. This can be explained by the size of the Brazilian domestic market, its crushing capacity and high domestic consumption. This suggested that this gave the market a buffer zone that smoothed the international price fluctuations in the short term. The global VEC model that included Rotterdam and Chicago as independent variables was found to be statistically significant, with an adjustment speed of the error correction term of 27.35% (3.5 months). This global model associated Chicago

with an increase in prices, explaining 15.11% of the increase, and Rotterdam with price decreases, explaining 12.14% of the decrease (Table 12).

**Table 11.** Vector error correction pairs model for Paranaguá.

| | Lag | ΔPARANAGUA | < | ΔCHIGF | Elasticity | Correlation LM Tests | Normality Test | Without Cross Term | With Cross Term |
|---|---|---|---|---|---|---|---|---|---|
| CointEq1 | 1 | −26.08% | | −7.35% | 1:0.94 | 0.1305 | 0.00020 | 0.0015 | 0.0023 |
| | lag | ΔPARANAGUA | < | ΔROTTERDAM | | | | | |
| CointEq1 | 1 | −27.32% | | −7.49% | 1:1.056 | 0.9821 | 0.00020 | 0.0002 | 0.0001 |

**Table 12.** Vector error correction global model for Paranaguá.

| Lag | ΔPARANAGUAS | < | ΔCHIGF | ΔROTTERDAM | | | | |
|---|---|---|---|---|---|---|---|---|
| 1 | −27.35% | | −15.11% | 12.14% | | | | |
| | | | | | Correlation LM | Normality Test | Without Cross Term | With Cross Term |
| | | | | | 0.79050 | 0.00000 | 0.0132 | 0.0010 |

Equation (10): VECM corrected using coefficient significance

$$\Delta LNPARANAGUASP = -0.260(LNPARANAGUASP_{t-1} - 0.942LNCHIGF_{t-1} - 0.403) + 0.410001\Delta LNPARANAGUASP_{t-1}$$
$$\leftarrow \textit{No short-term regressors from Chicago}$$
$$\lambda = -0.260808$$

(10)

Equation (11): VECM corrected using coefficient significance

$$\Delta LNPARANAGUASP = -0.273(LNPARANAGUASP_{t-1} - 1.05628324967LNUSROTTERDAMCIF_{t-1}$$
$$+ 0.399379675966) + 0.422625\Delta LNPARANAGUASP_{t-1}$$
$$\leftarrow \textit{No short-term regressors from Rotterdam}$$
$$\lambda = -0.273239$$

(11)

Equation (12): VECM corrected using coefficient significance

$$\Delta LNPARANAGUASP = -0.273(LNPARANAGUASP_{t-1} - 0.915LNCHIGF_{t-1} - 0.0366NUSROTTERDAMCIF_{t-1}$$
$$- 0.341) + 0.393\Delta LNPARANAGUASP_{t-1}$$
$$\leftarrow \textit{No short-term regressors from Chicago Futures and Paranaguá Futures}$$

(12)

### 3.5.4. VECM Rosario Spot

The VEC model for the Rosario Spot market showed a moderate market efficiency, as represented by the adjustment speeds of the error correction terms: −0.213640 (λ) and −0.2530 (λ) for Chicago and Rotterdam, respectively; any shocks in those markets and Rosario returned to the long-term equilibrium in around four and five months, respectively (Table 13). However, the fastest speed of adjustment occurred between Paranaguá and Rosario (λ = −0.3186) and had a lag period of 3 months before returning to the long-term equilibrium, where the first market was associated with 4.49% of the price increases in Rosario (Table 13). The results became counterintuitive when the VEC model for Rosario Spot explained by Rosario Futures showed a lower adjustment speed (20% per period) than other leading markets (Paranaguá, Chicago and Rotterdam). However, the model indicated that Rosario Futures could explain and was associated with almost half of the price decreases (47%) in Rosario Futures, while Rotterdam and Paranaguá could be associated with 16.91% and 5.63% of the price decreases in Rosario Futures due to Chicago and with associated price increases explaining 4.95% (Table 13). The construction of a global model was only significant at an alpha of 10%, where this model showed the fastest price transmission (λ = −0.40) and significant short-term and long-term regressors on both sides of the equations (Equation (17)). In this overall model, Rosario Futures was the most

important market associated with explaining the price decreases at the first lag $(t − 1)$, followed by Rotterdam $(t − 1)$. Despite the interesting result of the overall model, the last one was not statistically significant at an alpha of 5% and the residuals were not normally distributed (Table 14).

**Table 13.** Vector error correction pairs model for Rosario Spot.

| | Lag | ΔROSSP | < | ΔCHIGF | Elasticity | Correlation LM Tests | Normality Test | Without Cross Term | With Cross Term |
|---|---|---|---|---|---|---|---|---|---|
| CointEq1 | 3 | −25.30% | | −4.95% | 1:0.798 | 0.4155 | 0.4298 | 0.0300 | 0.0591 |
| | lag | ΔROSSP | < | ΔROTTERDAM | | | | | |
| CointEq1 | 1 | −21.36% | | 16.91% | 1:0.848 | 0.3810 | 0.00220 | 0.1708 | 0.2306 |
| | lag | ΔROSSP | < | ΔPARANAGUAS | | | | | |
| CointEq1 | 3 | −31.86% | | 5.63% | 1:0.858 | 0.3605 | 0.01930 | 0.4502 | 0.4405 |
| | lag | ΔROSSP | < | ΔROSFT | | | | | |
| CointEq1 | 1 | −20.37% | | 46.99% | 1:1.149 | 0.5674 | 0.0720 | 0.0005 | 0.0007 |

**Table 14.** Vector error correction global model for Rosario Spot.

| Lag | ΔROSSP | < | ΔCHIGF | ΔROTTERDAM | ΔPARANAGUAS | ΔROSFT | | |
|---|---|---|---|---|---|---|---|---|
| 3 | −40.53% | | −27.81% | 37.29% | −4.64% | 53.63% | | |
| | | | | Correlation LM | Normality Test | Without Cross Term | | |
| | | | | 0.1255 | 0.00000 | 0.2217 | | |

Equation (13): VECM corrected using coefficient significance

$$\Delta \text{LNROSSP} = -0.253(\text{LNROSSP}_{t-1} - 0.798\text{LNCHIGF}_{t-1} - 0.852) + 0.424\Delta\text{LNROSSP}_{t-1}$$
$$+ 0.273\Delta\text{LNROSSP}_{t-3} - 0.293\Delta\text{LNCHIGF}_{t-3} \tag{13}$$
$$\lambda = -0.253019$$

Equation (14): VECM corrected using coefficient significance

$$\Delta \text{LNROSSP} = -0.213(\text{LNROSSP}_{t-1} - 0.848\text{LNUSROTTERDAMCIF}_{t-1} - 0.456) + 0.217\Delta\text{LNROSSP}_{t-1}$$
$$\leftarrow \textit{No short-term regressors from Rotterdam (alpha 10\%)} \tag{14}$$
$$\lambda = -0.213640$$

Equation (15): VECM corrected using coefficient significance

$$\Delta \text{LNROSSP} = -0.318(\text{LNROSSP}_{t-1} - 0.858 * \text{LNPARANAGUASP}_{t-1} - 0.443) + 0.388\Delta\text{LNROSSP}_{t-1}$$
$$\leftarrow \textit{No short-term regressors from Paranaguá Spot} \tag{15}$$
$$\lambda = -0.318580$$

Equation (16): VECM corrected using coefficient significance

$$\Delta \text{LNROSSP} = -0.203(\text{LNROSSP}_{t-1} - 1.149 * \text{LNROSFT}_{t-1} + 0.823) + 0.470\Delta\text{LNROSFT}_{t-1}$$
$$\leftarrow \textit{No short-term regressors from Rosario Futures} \tag{16}$$
$$\lambda = -0.203718$$

Equation (17): VECM corrected using coefficient significance

$$\Delta \text{LNROSSP} = -0.405(\text{LNROSSP}_{t-1} + 0.373 * \text{LNCHIGF}_{t-1} + 0.096\text{LNUSROTTERDAMCIF}_{t-1}$$
$$- 1.015\text{LNPARANAGUASP}_{t-1} - 0.466 * \text{LNROSFT}_{t-1} + 0.295) - 0.28\Delta\text{LNCHIGF}_{t-1} - 0.283\Delta\text{LNCHIGF}_{t-3}$$
$$+ 0.372\Delta\text{LNUSROTTERDAMCIF}_{t-1} - 0.452\Delta\text{LNPARANAGUASP}_{t-2} + 0.536\Delta\text{LNROSFT}_{t-1} \text{ (alpha 10\%)} \tag{17}$$
$$\lambda = -0.405295$$

## 4. Discussion

This study was comprehended within the framework of a comprehensive study of the efficiency and dynamicity of the soybean market and how the international market

confronted the geopolitical situation (US–China trade war). The unit root test and the Bai–Perron test with multiple breaks were useful to scale and downsize the trade war in comparison with other historical events that had affected the soybean market. The conflict showed that only a single structural break found could be partially associated with the US–China trade war. This suggested that the international market has arbitraged around the Chinese tariff, neutralising the price dislocation effect. This might be associated with a highly efficient and fully cointegrated market. The first attempt of this research showed that the international markets were highly cointegrated, except for Rosario Futures, which was only cointegrated with the Rosario Spot market. Against expectations, the China Spot market presented a high number of cointegration equations with the other markets; this finding is in contrast with what is normal for government-intervened markets. The Chinese government has systematically intervened to shield the market from the price volatility of the international market, such as the "price minimum" initiative, "target price" and "deficiency payment policies" (Jamet and Chaumet 2016). This intervention provoked partial price dislocation within the international market, at least in the short term. Vavra and Goodwin (2005) pointed out that import tariffs will allow, in relative terms, full price transmission; hence a proportional price increase in international prices will generate a proportional increase in the domestic market price, except for prohibitively high tariffs. Consequently, prices in relative terms are fully transmitted. It is possible to assume that this was not the case for China, as the different combinations of price interventions shielded the market from international price volatility. The last empirical fact can be supported by the lack of short-term significant regressors in the short-term equation. In Argentina's case, it is well known for having an intervened domestic market; the Argentinian government designed different intervention policies, such as export tariffs and currency interventions (Margarido et al. 2001; Vassallo et al. 2011). All these policies distort price transmission and can generate asymmetrical price transmission and price dislocation. However, for the Argentinian case, the tariff may act as a fixed cost, enabling price transmission. The Brazilian market, represented by the Paranaguá market, presented cointegration with Chicago and Rotterdam. This falls in line with previous research by Margarido and Sousa (1998) and Mafioletti (2001).

The lack of cointegration vectors for many of the results in the first attempt of finding cointegration can be explained by a long period of market instability where several breaks occurred. All the breaks were associated with different exogenous shocks that were generated by shortages or surpluses, a difference in stocks and market speculation. These shocks had different origins, such as climatic problems and government interventions that affected the market, introducing breaks in the time series. The second attempt to find cointegration between the time series using the previously identified structural breaks as dummy variables was found to be effective by finding cointegration between all the main international markets. In other words, all series followed a long-term equilibrium, and this finding can be considered unique as many authors had shown that the soybean markets were integrated to some degree: USA (Correia das Neves 1993; Pino and Rocha 1994; Lima and Burnquist 1997; Margarido and Sousa 1998; Mafioletti 2001; de Moraes 2002; Giembinsky and Holland 2003; Da Silva et al. 2005); Brazil and Rotterdam (Europe) (Margarido et al. 2001); for Brazil, Rotterdam and Argentina (Margarido 2012); Argentina, Brazil, Chicago (USA) and Rotterdam (Europe) (Margarido et al. 1999; Machado and Margarido 2000; Da Silva et al. 2005; Margarido et al. 2007). Furthermore, cointegration was found between the Chicago Prices.

The vector error correction models using pairs and the global approach resulted in a suitable methodology to understand the dynamics of the international market. The first results showed a lack of short-term regressors in several models, indicating the strengthening of the domestic market and industry, such as in the case of the Brazilian market of Paranaguá. The other explanation involving statistically insignificant short-term regressors was government intervention, such as in the case of the China Spot market. Despite the structural government intervention in the domestic market, the China Spot market presented significant long-term regressors (at an alpha of 5%), meaning that Chinese author-

ities considered the Chicago prices as a reference for fixing the domestic price (Jamet and Chaumet 2016). China presented only a long-run causal relationship with other markets and a very slow speed of adjustment with all the different markets. These facts showed an incomplete price transmission and probably asymmetrical price transmission. In the short term, all international prices were frankly exogenous from China's domestic price system, and in the long term, almost all markets presented a long-term causal relationship with all the studied markets. Dalian Futures showed a consistently higher market efficiency, as represented by the speed of adjustment and higher market cointegration (Rapsomanikis et al. 2013).

The Paranaguá market showed a very high speed of adjustment from shocks in Chicago and Rotterdam. Rotterdam seemed to lead this market in the long term. In the short term, the VEC model using pairs did not present statistically significant coefficients, showing that the selected market did not have a short-term causal relationship with Paranaguá. This finding was in contrast to Margarido et al. (2001), where the difference in findings can be explained by the period that both studies covered and the current development of the Brazilian domestic market and significant crushing industry. This industry has a high demand for soybean oil from the Brazilian domestic market. This heavy demand for national soybean can buffer short-term price fluctuations. The Rosario Spot market showed relatively efficient behaviour in the pair model with Paranaguá, adjusting at rates of 31.86% per period from Paranaguá shocks. Along the same line, Rosario adjusted at 26.80%, 27.27% and 20.37% per period for Chicago, Rotterdam and Rosario Futures, respectively. The Argentinian market was systematically intervened in by the government to discourage soybean exportation and encourage the development of the crushing industry (Margarido et al. 2007; Vassallo et al. 2011). Despite this market intervention, the market efficiency marginally increased from the last empirical research performed by Margarido et al. (2001), when the speed of adjustment was found to be 26.16%. Margarido (2012) studied the relationship between the Brazilian and Argentinian markets, finding that prices in Argentina were not transmitted to Brazilian prices, which was consistent with this research's findings. In other words, Argentina (Rosario Spot) did not present short-term and long-term causalities with Brazil (Paranaguá). However, the results of this research showed that there was long-term and short-term causal relations between Brazilian prices and those of Argentina (Brazilian prices affected Argentinian prices). Rosario Futures affected the Rosario Spot market via Granger causality. However, Rosario Futures market did not present a short-term statically significant regressor (VECM) for Rosario Spot. The effect of Rosario Futures could explain almost half of the price decreases in Rosario Spot. To some degree, the Argentinian government intervention failed to dislocate the market; the tariffs acted as fixed costs and the increase in international price provoked a proportional increase in domestic market.

In summary, this research failed to reject the null hypothesis that the trade war dislocated the market; there was not enough empirical evidence to support the claim that the trade war provoked breaks in Chicago, Brazil, Rotterdam and Argentina. Therefore, it was not possible to use the structural break correlated with the trade war as a dummy variable to find cointegration. This finding reduced the significance of the trade war and it is necessary to reformulate the question and understand why the US–China trade war did not affect the international market, as well as other events over the past 10 years have. How the international market overcame the tariff and arbitrage around it can be partially explained by the results of the vector error model that showed a highly efficient market that was capable of efficiently arbitraging around the tariffs with the fast speed of transmission and a highly cointegrated market.

## 5. Conclusions

In general, the international market showed itself as highly efficient and cointegrated with a fast speed of adjustment from exogenous shocks in the leading markets. Chicago remained the most important market, leading the international prices. In other words,

Chicago affected all prices, except for the Rotterdam price. However, both Chicago and Rotterdam failed to present short-term, causal relationships with most of the markets. This can be explained by the development of the domestic market and crushing industry, as well as the domestic and international demand for soybean derivates (soybean oil, soymeal, etc.) in those markets. The development of domestic demand acted as a buffer zone for international price fluctuation, at least in the short-term (Brazilian case). The next scenario of lack of short-term causal relationship was explained by government interventions, such as in the case of China, where the market was price dislocated, at least in the short term. Rotterdam continued to be the second-most important centre of price formation, presenting a long-term causal relation with all markets, excluding Chicago. The Brazilian market rose to a significant level, presenting a long-term causal relationship with markets such as Rosario Spot, China Spot and Dalian Futures. Paranaguá rose as a leading market, creating the following question: to what extent did the trade war speed up this process? For a market with high market efficiency and market cointegration, the consequences of the trade war seemed to be relatively less significant. First, the tariff imposed by China did not cause a structural break in the main markets (Chicago, Rotterdam, Paranaguá and Rosario); therefore, price dislocation did not happen, at least in the short term. The tariff acted as a fixed cost for the Chinese importation of US soybean. The four markets managed to arbitrage around the tariff and return to the long-term price equilibrium. This is an excellent example of how free markets that are highly efficient regulate themselves, managing to overcome government interventions.

### 5.1. Policy Implications

The empirical evidence suggested the extraordinary capability of free markets to reorganise and arbitrage around tariffs, where the LOOP redirected the commercial flow around the tariff, equalising the international prices. The market agents acted fast to exploding arbitrage opportunity and the market converge under the same price. In terms of policy implication, China underestimated the market capability of the soybean market to rearrange and miscalculated the real impact of a tariff on US soybean. On the other hand, the Trump administration acted demagogically by giving farmers subsidies and distorting the price. The implementation of subsidies was unnecessary because the prices had converged fast due to the already explained arbitrage process. The trade war showed how well and efficient free markets work, making this event irrelevant and statistically not significant. The soybean trade war was one of several battles fought by the largest two economies of the world. According to Bown and Kolb (2020), this war generated a loss of economic welfare for the final consumer.

### 5.2. Recommendation for Further Study

This research quantified and described the market relation of the different international markets of soybean. Future research should focus on the individual contribution of each variable in the VECM by utilising the forecast error variance decomposition (FEVD). This will quantify the degree of explanation or contribution of each independent variable to a dependent in the autoregression model. Moreover, utilising the IRF will determine how the model variables respond to external shocks in one or more variables and give the model reaction as a function of time. This will help to identify the market power and build an overall understanding of the dynamic behaviour market by contrasting the magnitudes of different shocks (financial crisis, droughts, etc.) with the shocks introduced by the trade war.

The determination of the asymmetric price transmission on government-intervened market can generate insights regarding policy implications and market power. Through the threshold autoregressive model (TAR), momentum threshold autoregressive model (M-TAR) and asymmetric error correction model (AECM), these techniques will help to evaluate the loss of market efficiency due to trade war government interventions. Furthermore, Phillis and Perron (1998) stated that the unit root test could be applied to improve the unit root detection, and the Engle and Granger (1987) cointegration test can be used to

contrast with the Johansen cointegration test. This will result in a complete quantification of market efficiency and the impacts of government interventions and policies on the selected markets.

Despite the fact that there is enough empirical evidence of the international market condition before the trade war, the methodology and the econometric tools applied might differ from the methodology used in this research. Therefore, to evaluate and compare results to assess the consequences of the trade war, it is necessary to investigate the market in the three different periods: before the trade war, during the trade war and after the trade war using the same methodology.

Finally, the rise of the Brazilian market is evident; therefore, it is important to evaluate the main component of this upsurge. As aforementioned, the trade war might have had some direct implications on this phenomenon.

**Author Contributions:** D.P. and K.B., supervision, project administration, methodology. G.B.M., conceptualization, formal analysis, investigation writing and review and editing. All authors have read and agreed to the published version of the manuscript.

**Funding:** This research received no external funding.

**Informed Consent Statement:** Not applicable.

**Data Availability Statement:** All data sources come from public data, available in the website of the followings institutions; the international monetary fund (IMF), Chicago Board of Trade (CBOT), CEPEA Brazil (Centro de Estudos Avançados em Economia Aplicada), Dalian Commodity exchange (China), Wind Economic Database (China), and Bolsa de Rosario (Argentina). Most time series can be found at World Bank website. World Bank: https://www.worldbank.org/en/research/commodity-markets (accessed on 1 May 2019). Bolsa de Comercio de Rosario: https://www.bcr.com.ar/es/mercados/mercado-de-granos/cotizaciones/cotizaciones-locales-0 (accessed on 1 May 2019). CEPEA: https://www.cepea.esalq.usp.br/br/indicador/soja.aspx (accessed on 1 May 2019).

**Acknowledgments:** The authors acknowledge the support given by Stephen Giles and Sarah Nickels.

**Conflicts of Interest:** The authors declare no conflict of interest.

## Appendix A

**Table A1.** Johansen cointegration test using breaks as dummy variables.

| Unrestricted Cointegration Rank Test (Trace) | | | | | | |
|---|---|---|---|---|---|---|
| **PAIR TIME SERIES** | | **No. of CE(s)** | **Eigenvalue** | **Statistic** | **Critical Value** | **Prob.** |
| | | Hypothesised | | Trace | 0.05 | |
| LNROSFT LNROTTERDAM | Break 2014M8 | None | 0.128 | 24.75 | 15.49 | 0 |
| | Trend assumption 3 | At most 1 | 0.066 | 8.19 | 3.84 | 0 |
| LNROSFT LNROSSP | Break 2015M10 | None | 0.125 | 22.73 | 15.49 | 0 |
| | Trend assumption 3 | At most 1 | 0.054 | 6.63 | 3.84 | 0.01 |
| LNPARANAGUASP LNROSFT | Break 2014M09 | None | 0.15 | 28.67 | 15.49 | 0 |
| | Trend assumption 3 | At most 1 | 0.073 | 9.11 | 3.84 | 0 |
| LNDALIF LNROSSP | Break 2017M10 | None | 0.124 | 22.5 | 15.49 | 0 |
| | Trend assumption 3 | At most 2 | 0.053 | 6.56 | 3.84 | 0.01 |
| LNDALIF LNROSFT | Break 2010M10 | None | 0.156 | 30.35 | 25.87 | 0.01 |
| | Trend assumption 4 | At most 1 | 0.079 | 9.91 | 12.51 | 0.13 |
| LNCHINASP LNROTTERDAM | Break 2010M10 | None | 0.099 | 13.22 | 12.32 | 0.03 |
| | Trend assumption 1 | At most 1 | 0.006 | 0.7 | 4.13 | 0.45 |
| LNCHINASP LNROSSP | Break 2010M10 | None | 0.099 | 12.87 | 12.32 | 0.04 |
| | Trend assumption 1 | At most 1 | 0.003 | 0.38 | 4.129 | 0.6 |
| LNCHINASP LNROSFT | Break 2010M10 | None | 0.082 | 13.35 | 15.49 | 0.1 |
| | Trend assumption 3 | At most 1 | 0.025 | 3.07 | 3.84 | 0.08 |
| LNCHINASP LNPARANAGUASP | Break 2014M04 | None | 0.129 | 20.2 | 15.49 | 0.01 |
| | Trend assumption 3 | At most 1 | 0.03 | 3.6 | 3.84 | 0.06 |
| LNCHIGF LNROSFT | Break 2010M10 | None | 0.166 | 32.25 | 25.87 | 0.01 |
| | Trend assumption 4 | At most 1 | 0.083 | 10.4 | 12.51 | 0.11 |
| LNCHIGF LNCHINASP | Break 2010M11 | None | 0.159 | 29.3 | 25.87 | 0.02 |
| | Trend assumption 3 | At most 1 | 0.068 | 8.51 | 12.51 | 0.21 |
| LNROSSP LNROTTERDAM | Break 2010M11 | None | 0.129 | 19.57 | 15.49 | 0.01 |
| | Trend assumption 3 | At most 1 | 0.024 | 2.95 | 3.84 | 0.09 |

The highlight colour represents the number of cointegration equations.

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
