# Peer review of "Price Transmission Analysis of the International Soybean Market in a Trade War Context"

_economies, doi:10.3390/economies10080203_

Round 1

Reviewer 1 Report

The topic of the paper is interesting, I congratulate the group of authors:

The main suggestions for changes are as follows:

Please select the type of manuscript (Line L1) – probably Article

Please remove the space line between L41 and L42. Also, L97, L349; L1039-1044;

Please remove the space line between: L133-134; L160-161; L174-175; L218-219; L238-239; L250-251; L257-258; and so on, till the end.

Please verify again if the paper

Please check again if the work corresponds to the template.

Author Response

Thank you very much for taking the time for reading the paper! 

I have re-check the documents correct several grammar issues and delete all the spaces within paragraphs 

Reviewer 2 Report

Dear Authors,

I enjoyed reading your manuscript. I found the research topic and questions very interesting and you did a great job introducing the problem, explaining basic concepts, and discussing the implications of the findings.

However, I had some trouble following your analysis and discussion of results. The manuscript needs to be read again carefully and edited to fix some grammar and spelling issues (there were quite a few English grammar errors and some unclear sentences, also spelling issues such as “alfa” instead of "alpha" to denote significance). But also, some details need to be added to improve overall clarity. There was also some inconsistency in terms of reported results – some tables and/or models appeared missing (e.g., global VECM for Dalian futures) and some notes under the tables would be helpful. Besides that, please see more comments below.

Line 25: Is something missing in “US battles”? US-led battles, or US-initiated battles?

Lines 46-47: Some citations would be helpful here.

Line 104: Domestic soybean market?

Line 105: International soybean price? It is also confusing to read that international soybean price at CBOT leads international prices – perhaps specify what international prices?

Line 270: What about adjustments for inflations, were these done as well?

Line 364: Are some hypotheses missing in point 2.1.2? There is only one listed in the manuscript.

Line 380: Specify the null hypothesis for what. Non-stationarity?

Line 383: Original price series or the log-transformed price series were differenced?

Lines 400-401: Which are those 17 breaks? Looking at the tables, it appears that there are more than 17. Also, it is not clear what the highlighted cells represent, are these the breaks? It would be helpful to mention that in the table notes.

Line 440: Leave a note about the meaning of 1s and 0s? Also, there is a missing value for the cointegration between LNCHINASP and ROTTERDAM. Also, LN is missing in some price names.

Lines 460-461: In table 7 for Granger Causality matrix, it would be helpful to include descriptions of what the symbols mean in the notes.

533-536: How were the percentages determined?

Line 549: Why are Dalian Futures not the most impactful market, given the coefficient for the error correction term is the largest among all pairs? Is it not statistically significant?

Line 604: Where is the 3% value coming from? It looks like it's missing in Table 10 and the list of equations.

Lines 605-606: The sentence is not clear, and how were the percentages calculated?

Lines 610-611: Why is the model discussed here not reported?

Line 620: Equation number missing.

Lines 708-709: Why is a model estimated between Rosario spot prices and Rosario futures prices, which appears to assume that Rosario futures prices affect Rosario spot prices, if the Granger causality test in table 7 indicates that Rosario futures do not affect Rosario spot prices?

Lines 710-714: The sentence is not clear. Also, how were the percentages determined?

Line 712: Rosario Futures or Spot?

829-833: Are these two sentences contradictory? Also, there appears to be a short-run effect of Chicago prices on Chinese prices (see equation 1).

Lines 784-786: I see that there are five VECM pairs that involved Chinese prices as the explained variable (section 3.5.1), so how is there “the lowest number of cointegration vectors with other markets”? Did authors consider statistical significance and magnitude of the error correction coefficients to conclude this?

Lines 836-837: is this sentence correct? “..in the long term almost all market presented a long-term causal relationship with all the studied market.”

Lines 863-864: “Rosario Spot market presented short term regressor (VECM) for Rosario Futures...” – is it not Rosario spot prices affected by Rosario futures?

Author Response

Many thanks for taking the time for reading the paper , thank you for spotting several mistakes 

  • Lines 610-611

The creation of a global model for Dalian incorporating all the cointegrated series resulted in a spurious regression even at alpha 10% , therefore has not been included in this investigation.

  • Line 784-833 . Mistakes spotted “cointegration equations” rather than cointegration equations. And actually, this statement was wrong due that the number of cointegration equations wasn’t low, it was the same as that Paranaguá (Johansen cointegration). Therefore, this has been corrected.
  • Line 533-536 , determined by the error correction term
  • Line 549, Yes! because it does not have significant short-term regressors
  • Lines 863-864 ). Rosario Spot market presented short-term regressor (VECM) for Rosario Futures, Is it not Rosario spot prices affected by Rosario futures? This has been corrected Rosario Spot is granger caused by Rosario Futures but the vecm model doesn’t not present statistically significant regressors for the short-term equation
  • Line 836-837 “is this sentence correct? the long term almost all markets presented a long-term causal relationship with all the studied markets.” Yes, from the granger causality test and for the VECM this can be concluded

  • Line 604, thank you for spotting the mistake! I have rewritten “correction; term fluctuated between 13.8% and 7.5%.”
  • 829-833: the short-term regressor found in equation number 1 was at alpha 10% therefore is not valid not significant for statistical convention

Other comments have been addressed in the paper and highlighted in green

Reviewer 3 Report

Dear Authors,

1. The topic is interesting, also from the point of view of problems of  price transmission of the food and agricultural products market, especially in a trade war context. Your paper explains the problem with the use of advanced econometric methods. This Article is interesting and well qualified in publication.

I recommend You some changes to take into consideration.

1. In the line 1. I think, that it is original research article, not “Review, Communication, etc.”

2.Figure 1. Can You  edit Figure 1. better? I hope the data is available and You can create a better chart.

3. In the line 620 – “Equation Error! No text of specified style in document.. VECM corrected by coefficient significance”. There is no number of Equation (Equation 7. ?). and “Error! No text of specified style in document”.  Please, correct it.

In spite of all these lines, I consider that the Authors have done good work. The topic of this paper is appropriate to the scope of Economies Journal. Finally, it is necessary to review all references and adopt the rules of the editorial. I recommend the article to be published.

Author Response

Thank you very much, for taking the time for reading the paper.

I have improved the format, addressed grammar issues and re-done the chart with the data following the format